# Interference Management with Reflective In-Band Full-Duplex NOMA for Secure 6G Wireless Communication Systems

**DOI:** 10.3390/s22072508

**Published:** 2022-03-25

**Authors:** Rabia Khan, Nyasha Tsiga, Rameez Asif

**Affiliations:** 1Power Networks Demonstration Centre, University of Strathclyde, Glasgow G1 1XH, UK; 2Mechanical Engineering, Moscow Automobile and Road Construction State Technical University (MADI), 125319 Moscow, Russia; tsiga.nyasha@yahoo.com; 3School of Computing Sciences, University of East Anglia, Norwich Research Park, Norwich NR4 7TJ, UK; rameez.asif@uea.ac.uk

**Keywords:** 6G, B5G, in-band full-duplex, reflective-in-band full-duplex, non-orthogonal multiple access, machine learning genetic algorithm, secrecy capacity

## Abstract

The electromagnetic spectrum is used as a medium for modern wireless communication. Most of the spectrum is being utilized by the existing communication system. For technological breakthroughs and fulfilling the demands of better utilization of such natural resources, a novel Reflective In-Band Full-Duplex (R-IBFD) cooperative communication scheme is proposed in this article that involves Full-Duplex (FD) and Non-Orthogonal Multiple Access (NOMA) technologies. The proposed R-IBFD provides efficient use of spectrum with better system parameters including Secrecy Outage Probability (SOP), throughput, data rate and secrecy capacity to fulfil the requirements of a smart city for 6th Generation (6thG or 6G). The proposed system targets the requirement of new algorithms that contribute towards better change and bring the technological revolution in the requirements of 6G. In this article, the proposed R-IBFD mainly contributes towards co-channel interference and security problem. The In-Band Full-Duplex mode devices face higher co-channel interference in between their own transmission and receiving antenna. R-IBFD minimizes the effect of such interference and assists in the security of a required wireless communication system. For a better understanding of the system contribution, the improvement of secrecy capacity and interference with R-IBFD is discussed with the help of SOP derivation, equations and simulation results. A machine learning genetic algorithm is one of the optimization tools which is being used to maximize the secrecy capacity.

## 1. Introduction

The progression of technology requires a smart way of enactment of the available state of the resources with the new and smart algorithms. Recently, 5G is implemented in metropolitan cities of the world using high-frequency ranges of the spectrum. High frequencies are considered harmful for the health and well-being [1]. Sophisticated use of the whole existing electromagnetic spectrum perhaps reduces the threats and risks for organisms. Several approaches have been investigated [2] by targeting the high-quality throughput with the existing proposed technologies like Full-Duplex (FD) and Non-Orthogonal Multiple Access (NOMA). FD and NOMA both focus on the efficient utilization of the spectrum for developing a system with better Spectral Efficiency (SE) by the thought-provoking way of spectrum exploitation. The current execution of 5G is mostly based on the Orthogonal Frequency Division Multiple Access (OFDMA) due to the high cost limitation for an entirely new NOMA based system. The 4G system is also based on OFDMA, therefore, the application of 5G on the existing 4G infrastructure is comparatively easier and cost-effective. Keeping the consideration of spectrum limitation, it will be better to extend the system further by using NOMA for the 5G onward technologies.

### 1.1. Related Work

The incorporation of NOMA with FD is a challenging yet promising approach for the 6G technology. The idea of combining NOMA with FD can improve the SE and the combination will also help in managing the drawbacks of each technology. Regarding the hardware progression in the propagation and digital and analogue domains [3], several studies have been put forward for the combination of both promising techniques. Table 1 shows the comparative study of the existing FD-NOMA literature.

The existing literature explores the advantages of FD and NOMA. For maintaining a balance between FD and Half-Duplex (HD) [8], the transmission switching mode from HD to FD and vice versa is one of the common approaches of communication. Beamforming is another successful approach for NOMA and FD. It can improve the data rate of a particular user with power allocation management [9]. It also provides a better ergodic sum-rate than HD [10]. However, the HD outperforms FD at high Signal-to-Noise-Ratio (SNR) in terms of outage probability and in the delay-tolerant transmission [8,10]. There are several approaches for successful implementation for NOMA and FD with high data rate [9] and ergodic sum-rate [10] including beamforming. Cognitive Radio-based NOMA (CR-NOMA) with femtocells is a good strategy for the deployment of NOMA using pairing techniques of the near and far users [11] and for the improvement of system overall performance.

A communication system is based on several layers, each layer has different requirements of security due to multiple connected objects in the beyond 5G [12]. Physical Layer Security (PLS) is one of the main concerns in a smart system. This literature focuses on the PLS. For PLS there are two types of eavesdropper/s (Eve) that cause threats to security; active and passive. It can be amongst the trusted intended user or an outsider. Several existing algorithms are obliging for tackling the issues of PLS, however, due to the advancement of technology, the security threats are also getting advanced. The enactment of technology everywhere including; banks, buildings, security systems, houses, schools, accounts, industries and transportation makes everything vulnerable. One of the possible ways of handling security issues of jamming and eavesdropping for a hybrid eavesdropping (passive eavesdropping on the transmission source and the reactive jamming on legitimate users) of FD-NOMA is by exhausting eavesdropper by increasing its decoding complexity [13]. After the development of a numerous algorithms [12,13,14], the security of a wireless communication system still needs improvement.

In the traditional communication system, each device is capable of dual communication with a transmission and a reception task called HD communication. In HD, the channel is shared between the dual tasks, with orthogonal (different) time slots or orthogonal frequency bands (out-of-band FD mode). FD allows a device to broadcasts and receives the signals in the same frequency band and time slot [15]. There are two types of FD: the Out-of-Band FD (OBFD) (which is a kind of HD) and the In-Band FD (IBFD) where the transmission and reception of the signal take place at the same time slot and frequency band. In terms of network capacity, system reliability, sum rate and other theoretical aspects IBFD is superior to conventional HD [3].

Three ways for the deployment of the IBFD are shown in Figure 1: (a) shows the combined NOMA downlink and uplink system where the transmitter and the recipient can simultaneously transmit and receive the signals, (b) shows the Device-to-Device (D2D) NOMA communication system, where the near user forwards the signal to the far user with and without a direct link between the near and far user, (c) shows the Two-Way Relay (TWR) NOMA communication system, where a relay assists the far user by receiving its signals from the transmitter. The relay forwards the previously received signal in the FD mode along with the reception of a new signal in the same time slot.

IBFD and D2D communication have common characteristics including better performance at short distances and reduced self-interference at low power transmit signals [3]. IBFD seems to be more spectrally efficient. Therefore, this literature focuses on the important problems, solutions, benefits, challenges and exploration in the proposed system with IBFD.

In IBFD mode, the wireless nodes cannot decode the signals easily. Characteristically, the transmitted signal is approximately 100 dB higher than the received signal which causes eroded Signal Interference (SI). This reduces its capacity below HD. According to consensus by both academia and industry, it is very difficult to achieve SI cancellation/suppression for IBFD integration [16]. The increasing interference cumulatively reduces the data rate, latency, secrecy capacity and throughput.

RF digital interference cancellation [3], advance antennas cancellation and digital base band, channel estimation and power allocation [4] offers the possible solutions for IBFD. Hybrid resources [17] that switch between FD and HD have been invented for developing the radio resources and simultaneously improving the SE. Separated resources are needed like antennas for the separate transmission with the minimum possible interference. The use of FD with NOMA is also helpful to prevent the spectral resources and the use of other existing devices to avoid system complexity. One solution is proposed in this article with R-IBFD. R-IBFD not only tackles the interference involved but also provide security due to inaccessible information to the Eve.

For the security improvement and confound the eavesdropper, a cooperative jammer [18] is used to analyse the outage performance of the system. A multi-point cooperative relay selection scheme [19], is atypical to the conventional cooperative communication, where each user assists the next further user for downloading its signal until the furthest user downloads its signal successfully. Authors used Channel State Information (CSI), Rayleigh fading and Nakagami-m fading channel for concluding the proposed approach. For reliability and security of a system with multi-antenna users and eavesdropper, single antenna source and Nakagami-m fading channel, an imbalance in-phase and quadrature component [20] is explored for its influence on the system. An Artificial Noise (*AN*)-aided cooperative communication scheme with relay firing the jamming signal [21] is proposed, where the ergodic secrecy sum-rate (ESSR) is used to evaluate the system secrecy performance.

For improving sum data rate, an opportunistic relay selection scheme [22] is employed and several wireless communication techniques are used to support the system. A resource allocation problem [23] for an interplay of NOMA, FD and D2D are proposed for sum-rate optimization where a group of strong users assist weak legitimate users.

IBFD can improve conventional wireless communication in terms of time delay, loss of data due to high congestion, hidden terminals and SE. This is to develop a heterogeneous dense network with high capacity and flexible relaying nodes. The quantitative analysis of theoretical and practical deployment shows that at the cost of increased complexity, FD shows high throughput, diversity, low symbol error rate and reduce the use of conventional HD. Alongside interference and security management, the proposed R-IBFD poses the above benefits. The large buffer size for FD devices can additionally reduce the Packet-Loss-Ratio (PLR) [24]. The D2D FD communication performance increases with the increase of in-cell communication ratio that leads to bandwidth efficiency [25]. The IBFD deployment needs new algorithms for the deployment of 6G. It aims to use high-frequency bands like millimetre waves (mmWaves). This is because the low-frequency spectrum is fully utilized with the existing below 5G systems.

To prevent the use of high frequencies, harmful for the health, it is better to reuse the spectrum with FD or NOMA for higher SE and better utilization of lower frequency bands.

NOMA offers a high bandwidth efficiency when implemented with IBFD. There are several types of NOMA. Categorically, power domain NOMA and code domain NOMA are the two major divisions. This paper focuses on the power domain NOMA with IBFD. This paper elaborates the problems, solutions, benefits and challenges offered for power domain NOMA.

In NOMA, the power is allocated with respect to the distance and the channel condition of the individual user, in the same frequency band. The user far from the transmitter needs the highest power allocation. The power allocation decreases with the distance and the improvement of the channel condition. The transmitter superposes signals of all users before transmission. The sum of all user’s power coefficients is always equal to 1. $\alpha_{1}+\cdots +\alpha_{N}=1$, where α is the coefficient of power allocation and N is the total number of user/s. At each receiving node, the nearest strong users need to do Successive Interference Cancellation (SIC) for all users’ signals that have power higher than its own signal. Similarly, other users decode their signals. The furthest weak user does not receive the high power signals of other users; therefore, it considers the power of others users with the better channel conditions as noise to decode its own signal [26].

Amongst many problems of NOMA, strong interference, SIC complexity and the use of high-power allocation are the major problems. Installation of the NOMA algorithm requires high computational power for both real-time power allocation and successive interference cancellation. The new technologies such as IoT, Multiple Input and Multiple Output (MIMO), high-speed web, interactive multimedia processing and TV streaming also require energy consumption. Normal battery life for integrating the technologies in 6G will not be sufficient.

There are a number of possible proposed solutions to overcome the interference and to maintain the balance in the system in terms of high-power allocation requirements. FD Cooperative NOMA (FD-CNOMA) shows better system fairness as compared to HD Cooperative NOMA (HD-CNOMA). However, the hybrid combination of switching between HD and FD communication is also a suitable way to maintain a balance between the interference problems [5]. Space-Time Block Coding NOMA (STBC-NOMA) is one of the approaches that are helpful for reducing SIC complexity in the system using Alamouti distributed STBC [27]. Modulation based NOMA (M-NOMA) is one of the techniques that offer reduced interference, less SIC complexity and is also energy efficient [28].

Depending on the available CSI, a singular value decomposition scheme [29] is proposed for solving security issues that arise using NOMA-MIMO implementation and secrecy rate analysis. The system analysis of a two-way cooperative relay NOMA communication [30] is observed with the joint effect of imperfect SIC and quadrature and in-phase imbalance and Rician fading channel. Performance degradation with an increased number of users is caused due to high SIC on each user’s end [31], to analyse the system symbol error probability is derived and simulated for imperfect SIC and Space-Time Block Code (STBC) aided NOMA with timing offsets.

A Matlab platform is designed [32] for system and link-level analysis and Long-Term Evolution (LTE) is used as a baseline technique for Bit Error Rate (BER) and capacity comparison. A two-hop relay selection NOMA for IoT communication system [33] is considered to enhance secrecy performance. Using a Tchebycheff approach [34], a system for a secure FD NOMA and SWIPT is studied with power minimization and multi-objective optimization problems.

To permit an effective Dynamic Power and Channel Allocation (DPCA) in the DL multi-channel NOMA (MC-NOMA) systems [26], the optimization as the combinative problem, and offers three investigative solutions, i.e., two-stage Greedy Randomized Adaptive Search (GRASP), stochastic algorithm, and two-stage Stochastic Sample greeDy (SSD). For shortage of frequency band resources, a Terrestrial-Satellite Integrated Network (TSIN) for ground users in [35] is evaluated with NOMA.

NOMA is an essential component for the upcoming 5G technology which will provide a promising SE. The allocation of subcarriers to the users of poor channel conditions affects the SE of the system in OMA. NOMA allows the use of the channel by all users together and offers user fairness, massive connectivity, low latency and diverse QoS. However, interference, system computational complexity and the required energy in NOMA still need thoughtful consideration. IoT requires the connection of everything, which seems very interesting but demands a lot. AI and ML together with FD and NOMA seem promising but challenging. Alongside, security is the biggest all-time issue. The security issue needs to be resolved more than the advancement of technology. The more connections amongst objects open, the more loopholes for security and privacy issues.

### 1.2. Motivation and Contributions

The requirement for innovative spectrum consumption, minimum interference and high security in a 6G wireless communication system motivated us to propose the R-IBFD algorithm. After a thorough study, it has been observed that co-channel interference is the main problem for enabling FD and NOMA in a communication system. A number of algorithms are given by different authors, however, due to the high connectivity of multiple devices, the security problem is not completely resolved. Keeping that in mind, the following are the contributions of this literature:A brief literature review for IBFD, NOMA and their system requirements in 6G technology for developing a background for the reader.Propose and discuss a novel Reflective In-Band Full-Duplex (R-IBFD) algorithm.Use R-IBFD for interference management and security enhancement.Derive secrecy outage probability for R-IBFD, for selection of a relay amongst K relays.Show the usefulness of the proposed R-IBFD with ML for the forthcoming 6G system with numerous devices and large data.Simulate R-IBFD for N-number of users to show minimal interference and security management as compared to baseline NOMA and HD.

This paper is an extension of a published paper in a conference [36] where the proposed algorithm is explored for a two-user system and verified the novelty using secrecy outage probability and throughput of the system.

## 2. Reflective In-Band Full-Duplex with NOMA in beyond 5G

The proposed R-IBFD aims to assist the relay in forwarding the signal in the IBFD mode. The proposed algorithm is explored for NOMA for improving the SE of the system with minimal interference and better security. For obtaining the benefits of IBFD and NOMA, a reflective IBFD (R-IBFD) cooperative communication algorithm is proposed. The R-IBFD supports the IBFD mode device to forward the message with cooperative communication, ideally without interference. It is slightly different from the existing Decode-and-Forward IBFD (DF-IBFD) cooperative communication. The user in the IBFD mode decodes its own signal by using SIC (real or imaginary) and encodes the signal that needs to be forwarded to an opposite component (real or imaginary) then forwards/reflects the signal to the distant user. The trusted IBFD mode user contains the CSI of other users. Before reflecting, the IBFD mode user adds *AN* for preventing the security issues. The use of R-IBFD reduces the decoding complexity and co-channel interference at the IBFD mode user.

### 2.1. System Model

Figure 2 shows the system model for R-IBFD with a D2D cooperative communication for a Downlink (DL) communication system. The proposed system model contains a transmitter (Alice), total *N* users (Bobs) and a passive Eve. Near user/s ( $Bob_{1}$, $Bob_{3}$, ⋯) and far users ($Bob_{2}$, $Bob_{4}$, ⋯, $Bob_{N}$ ) are considered to have similar channel conditions. Alice, Bobs and Eve contains a single antenna excluding $Bob_{1}$. Alice uses $P_{T}$ as the total transmission power. Power domain NOMA is used for the allocation of power to each users’ signal. $\alpha_{1}$, $\alpha_{2}$, $\alpha_{3}$, $\alpha_{4}$, ·, $\alpha_{N}$ where $\alpha_{1}+\alpha_{2}+\alpha_{3}+\alpha_{4}+\cdots +\alpha_{N}=1$. The power coefficients are allocated with respect to the distances $d_{1}<d_{3}<d_{2}<d_{4}<\cdots <d_{N}$ and the channel power gain $g_{4}\approx g_{2}<g_{3}\approx g_{1}$ of each Bob from Alice, statistically and with machine learning. As per consideration, each Bob is facing a Rayleigh fading channel and their CSI are known by Alice and $Bob_{1}$. Eve does not have CSI information of the legitimate user/s. The Eve’s channel is considered as worse than two near users conditions; therefore, Eve is unable to detect their signals. $g_{4}\approx g_{2}<g_{e}<g_{3}<g_{1}$. Eve is capable of detecting the signal of $Bob_{2}$ and $Bob_{4}$. The other channels included in the transmission are $g_{12}$ and $g_{11}$ are the channels for the R-IBFD communication in the IBFD mode, from $Bob_{1}$ to $Bob_{2}$ and the SI channel from $Bob_{1}$ to itself.

Alice modulates road 1, Bobs’ signal on the real component, and road 2, Bobs’ signal on the quadrature component of the 16-QAM modulation constellation. This is because real and quadrature components do not interfere with each other, therefore Bobs’ signals of road 1 do not interfere with Bobs’ signals of road 2. It assists in reducing interference as compared to NOMA signals.

$Bob_{1}$ and $Bob_{4}$ on road 1 are modulated on the real component and the remaining Bobs of road 2, $Bob_{2}$ and $Bob_{3}$, are modulated on the quadrature components of the modulation. Therefore, both Bobs of road 1 face the co-channel interference with each other only, as shown in Figure 2. Ideally, there is no interference between the Bobs of different roads. This is to avoid the interference between two high power signals (far users $Bob_{2}$ and $Bob_{4}$) with each other. Therefore, far users ideally do not interfere with each other, rather faces minor interference due to very low power signal, which can be considered as noise. According to the system model, the near IBFD mode user $Bob_{1}$ receives high power signals of far users therefore it can assist far user $Bob_{2}$ with R-IBFD. To decode its signal, $Bob_{1}$ performs SIC by decoding $Bob_{4}$’s signal and removing the combined *AN*
$A_{1}$ of $Bob_{2}$ and $Bob_{4}$, then subtracting it from the total received signal. $Bob_{1}$ decodes its own signal from the total received signal and adds *AN*
$A_{2}$, null space of $g_{12}$, forward the signal of $Bob_{2}$ and $Bob_{3}$, using R-IBFD, like a reflector, to assist $Bob_{2}$. $Bob_{3}$ cannot decode the signal received from $Bob_{1}$ due to the addition of $A_{2}$ which is a null space of $Bob_{2}$’s channel. For forwarding the signal, $Bob_{1}$ uses the power $P_{c}$. $Bob_{1}$ receives its signal, which was modulated on real component, it adds *AN* ($A_{2}$) for $Bob_{2}$ and forwards the signal of $Bob_{2}$ which was modulated on quadrature component as received from Alice. There is self-interference at the IBFD mode, between the complex component of the received signal by $Bob_{1}$ and the complex signal forwarded by $Bob_{1}$; however, this interference is less than the usual IBFD due to no real component interference.

### 2.2. Addition of Artificial Noise for Improved Security

*AN* is a sufficient way for the protection of transmission signals from Eve and other users. System design with *AN* depends on the receiver’s channel but not Eve’s channel. It is generated before transmission of a signal by the Alice and $Bob_{1}$ to degrade Eve’s channel. Both the signal *x* and the *AN*, $A_{n}$, are complex Gaussian in nature. In case of the fixed *AN*, the value of $\left|\right|g_{e}A_{n}\left|\right|$ might be smaller. To avoid this situation the value of *AN* is considered as the Gaussian random variable in the null space of $g_{n}$ of the Bob’s channels respectively, such that $g_{n}A_{n}$ is 0 [14].

## 3. System Analysis for a Two-User System

For the analytical insight of the proposed technique, two users are used to avoid system’s complexity. For a two user system, the same situation (Figure 2) is considered with $Bob_{1}$ and $Bob_{2}$ only. The superposed signal by the Alice for the broadcast can be given as: $s_{1}\left[t\right]=\sqrt{g_{1}P_{T}}\left(\sqrt{\alpha_{1}}x_{1}\left[t\right]+\sqrt{\alpha_{2}}x_{2}\left[t\right]\right)+A_{1}$. Likewise, $Bob_{1}$ add $A_{2}$ before forwarding the signal of $Bob_{2}$ as $s_{2}\left[t\right]=x_{2}\left[t\right]\hat{j}+A_{2}$.

The signal received by $Bob_{1}$ also includes the self-interference for its co-channel transmission as given in (2) of [37]:(1)$$\begin{array}{cc} y_{1}\left[t\right] & =K_{1}{d_{0}}^{\gamma }{d_{1}}^{-\gamma }\left[\sqrt{g_{1}P_{T}}\left(\sqrt{\alpha_{1}}x_{1}\left[t\right]+\sqrt{\alpha_{2}}x_{2}\left[t\right]\right)\right]\\ & +\sqrt{g_{11}\left[t\right]P_{c}}s\left[t\right]\hat{j}+w_{1}\left[t\right], \end{array}$$
where $w_{n}∼CN\left(0,\sigma^{2}\right)$ is the additive white Gaussian noise (AWGN) and s[t] is the signal transmitted by $Bob_{1}$ (received along with the previous transmission) to $Bob_{2}$ in the IBFD mode, which causes self-interference, $K_{n}$ is the path loss factor for node *n*, $d_{0}$ is reference distance, $P_{c}$ is the power allocated by $Bob_{1}$ and γ is the path loss exponent.

The total received signal at $Bob_{2}$ from $Bob_{1}$ is given below according to (5) in [37]:(2)$$y_{12}\left[t\right]=K_{12}{d_{0}}^{\gamma }{d_{12}}^{-\gamma }\sqrt{g_{12}P_{c}}x_{2}\hat{j}\left[t-\tau \right]+w_{2}\left[t\right].$$

In the above equation, the signal received from $Bob_{1}$ contains only the real part, as $g_{12}A_{2}=0$. The IBFD cooperative communication is used for $Bob_{2}$; therefore, without the loss of generality a delay τ has been introduced.

Eve’s received signal can be determined by adding *AN* in (Equation 2). Eve receives a signal with high interference due to a lack of information about AN and the modulation alteration used.
(3)$$y_{e}\left[t\right]=K_{1e}{d_{0}}^{\gamma }{d_{1e}}^{-\gamma }g_{1e}\left[t\right]\left(\sqrt{P_{c}}x_{2}\left[t-\tau \right]\hat{j}+A_{2}\right)+w_{e}\left[t\right].$$

### 3.1. Performance Evaluation

For the evaluation of a system capacity, secrecy capacity, secrecy outage probability and throughput are some of the important parameters to prove the authenticity and usefulness of any system. In R-IBFD, the source modulates users’ signal on the real component of the 4-QPSK constellation mapping and add complex AN to make a complex transmission signal. Selected near-user $Bob_{1}$, amongst K users, forwards the signal of the far-user after adding AN for the R-IBFD cooperative communication. For the selection of relay amongst K relay, a relay selection method is used which is described later in this section.

### 3.2. Computation of Secrecy Capacity

Each node received a certain level of SINR or SNR depending on its channel condition and the interference. In this paper, SINR and SNR will be used interchangeably and is denoted as ζ. The respective received ζ of the $Bob_{2}$ and Eve are given as:(4)$$\zeta_{2}=\min\left\{\frac{\zeta_{a1}}{\zeta_{11}+1},\zeta_{12}\right\}$$
and
(5)$$\zeta_{e}=\zeta_{re},$$
where $\zeta_{a1}={\frac{\alpha_{1}P_{T}}{G}}_{a1}\sigma^{2}$, $\zeta_{11}=\frac{P_{c}G_{11}}{\sigma^{2}}$, $\zeta_{12}={\frac{\alpha_{2}P_{T}}{G}}_{12}\sigma^{2}$ and $\zeta_{1e}={\frac{\alpha_{2}P_{T}}{G}}_{1e}A_{2}G_{1e}+\sigma^{2}$ which follows the exponential distribution with parameter $\lambda_{a1}={\frac{\alpha_{1}P_{T}}{G}}_{a1}\sigma^{2}$, $\lambda_{11}=\frac{a_{2}G_{11}}{\sigma^{2}}$, $\lambda_{12}={\frac{\alpha_{2}P_{T}}{G}}_{12}\sigma^{2}$, $\lambda_{1e}={\frac{\alpha_{2}P_{T}}{G}}_{1e}A_{2}+\sigma^{2}$ and $G_{ni}=K_{ni}{d_{0}}^{\gamma }{d_{ni}}^{-\gamma }g_{ni}$. For $\lambda_{1e}$, it is assumed that $\sigma^{2}=\sigma^{2}/G_{1e}$. The subscripts 1, 2, 11, 12, a1 and 1e show the parameters for $Bob_{1}$$Bob_{2}$, between $Bob_{1}$ and itself, $Bob_{1}$ and $Bob_{2}$, Alice and $Bob_{1}$, $Bob_{1}$ and Eve.

The achievable data rate for the $Bob_{2}$ and the Eve is given as:(6)$$R_{2}=\log_{2}\left(1+\min\left\{\frac{\zeta_{a1}}{\zeta_{11}+1},\zeta_{12}\right\}\right),$$
and
(7)$$R_{1e}=\log_{2}\left(1+\zeta_{1e}\right).$$

The possible secrecy capacity of the system for R-IBFD system is given as
(8)$$C_{sec}=\max\left\{0,R_{2}-R_{1e}\right\},$$
(9)$$C_{sec}=\max\left\{0,\log_{2}\left(\frac{1+\min\left\{\frac{\zeta_{a1}}{\zeta_{11}+1},\zeta_{12}\right\}}{1+\zeta_{1e}}\right)\right\}.$$

### 3.3. Relay Selection

For better secrecy performance in the presence of Eve an opportunistic relay selection scheme is used [37]. The scheme is based on the selection of the relay amongst *K* relays that maximize the secrecy capacity of the system
(10)$$R_{s}=\arg\max_{k=1,\;\cdots ,K}\left[\frac{1+\min\left\{\frac{\zeta_{a1}}{\zeta_{11}+1},\zeta_{12}\right\}}{1+\zeta_{1e}}\right].$$
where $R_{s}$ is the selected user ($Bob_{1}$). Whilst selecting the user to relay the signal, the relay selection scheme is considering the channel between near users and the Eve.

A centralized approach is used in this paper, where the source or destination keeps record of the *K* relays and their CSI. Using the criteria of (Equation 10), the best relay for the transmission is decided.

### 3.4. Computation of Secrecy Outage Probability

For the derivation of proposed system’s Secrecy Outage Probability (SOP), the min-max approached is used. The SOP for R-IBFD cooperative communication for the relay selection scheme is given as:(11)$$\begin{array}{cc} S_{op} & =Pr[C_{sec}^{R_{s}}<C_{th}]\\ \; & =Pr\left[\log_{2}\left(\frac{1+\min\left\{\frac{\zeta_{a1}}{\zeta_{11}+1},\zeta_{12}\right\}}{1+\zeta_{1e}}\right)<C_{th}\right],\\ \; & =\prod_{k=1}^{K}\int_{0}^{\infty }Pr\left[\log_{2}\left(1+\min\left\{\frac{\zeta_{a1}}{\zeta_{11}+1},\zeta_{12}\right\}\right)<a+by\right]\\ \; & f_{\zeta_{1e}}\left(y\right)dy,\\ \; & =\prod_{k=1}^{K}\int_{0}^{\infty }F_{Z}\left(a+by\right)f_{\zeta_{1e}}\left(y\right)dy. \end{array}$$
where Pr[.], $f_{X}\left(.\right)$ and $F_{X}\left(.\right)$ are the notation for probability, Probability Density Function (PDF) and Cumulative Distributive Function (CDF). $a=2^{C_{th}}-1$, $b=2^{C_{th}}$, $y=\gamma_{1e}\geq 0$ and $f_{\zeta_{1e}}=e^{-\frac{y}{\lambda_{1e}}}/\lambda_{1e}$. The CDF $F_{Z}\left(z\right)$ of the random variable *Z* is derived in Appendix A and is given as
(12)$$F_{Z}\left(z\right)=1-\frac{\lambda_{a1}}{\lambda_{a1}+\lambda_{11}z}e^{-z(\frac{1}{\lambda_{a1}}+\frac{1}{\lambda_{12}})}.$$

Substituting the required parameter and considering $\zeta_{a1}=\zeta_{12}$ the SOP is given as
(13)$$\begin{array}{cc} S_{op} & =\prod_{k=1}^{K}\int_{0}^{\infty }1-\frac{\lambda_{a1}}{\lambda_{a1}+\lambda_{11}z}e^{-z(\frac{1}{\lambda_{a1}}+\frac{1}{\lambda_{12}})}\frac{e^{-\frac{y}{\lambda_{1e}}}}{\lambda_{1e}}dy.\\ \; & =(\frac{e^{-\frac{2a}{\zeta_{a1}}}}{b\zeta_{1e}\zeta_{11}}(be^{\frac{2a}{\zeta_{a1}}}\zeta_{1e}\zeta_{11}+e^{L}\Gamma \zeta_{a1}+e^{L}\zeta_{a1}\log\\ & \left(\frac{1}{\zeta_{1e}}+\frac{2b}{\zeta_{a1}}\right)-e^{L}\zeta_{a1}\log\left(\frac{b\zeta_{11}}{a\zeta_{11}+\zeta_{a1}}\right)-\zeta_{a1}\\ & 1F1^{\left(1,0,0\right)}\left[1,1,L\right]){)}^{K}. \end{array}$$

The above expression is the conditional expression with Re[p]>0 and Re$[1/\zeta_{1e}+2b/\zeta_{a1}]\geq 0$ where $L=\frac{\left(2b\zeta_{1e}+\zeta_{a1}\right)\left(a\zeta_{11}+\zeta_{a1}\right)}{b\zeta_{1e}\zeta_{11}\zeta_{a1}}$, Γ= EulerGamma and 1F1[1,0,0] is the Kummer confluent Hypergeometric function. Wolfram Mathematica software is used for the derivations of this paper.

### 3.5. Secrecy Throughput Evaluation

Throughput is another significant system parameter that can clarify the authenticity of a system. The throughput in an R-IBFD system, when the relay uses its internal power for the transmission of $Bob_{2}$’s signal is given as
(14)$$\mathrm{TP}=C_{th}\left(1-S_{op}\right).$$

## 4. System Analysis for a Four User System and Machine Learning

The current and forthcoming demand of high data and large number of systems will increase the processing power and system complexity. For such situation, an MLGA is used for the optimization instead of an statistical optimization technique. The superposed signal by the Alice for the broadcast can be given as:(15)$$s\left[t\right]=s_{1}\left[t\right]+s_{2}\left[t\right],$$
where $s_{1}\left[t\right]=\sqrt{P_{T}}\left(\sqrt{\alpha_{1}}x_{1}\left[t\right]+\hat{j}\sqrt{\alpha_{2}}x_{2}\left[t\right]\right)$ and $s_{2}\left[t\right]=\sqrt{P_{T}}\left(\sqrt{\alpha_{3}}x_{3}\left[t\right]+\hat{j}\sqrt{\alpha_{4}}x_{4}\left[t\right]\right)+A_{1}$.

In this article, each user individually does not have a complex nature. However, it is combined with the signal of another user or/and *AN* to make the total complex signal. Here, two users are considered to have the same channel conditions, so that we can collectively add the same *AN* for two users, it will save the bandwidth and the data rate will remain higher. Since the allocation of *AN* to each user separately needs more bandwidth. Additionally, in this case, adding *AN* for the near users is not necessary as the Eve cannot decode their signal.

$Bob_{3}$ performs SIC to remove $Bob_{2}$’s high power signal as they are encoded on the same component. $Bob_{4}$ decodes the signal by considering $Bob_{1}$’s low power signal as noise. $Bob_{3}$ cannot decode the signal received from $Bob_{1}$, as $Bob_{1}$ has added $A_{2}$, that is the null space of the channel $g_{12}$ of $Bob_{2}$ only. The signal received by the $Bob_{1}$ also includes the self-interference for its co-channel transmission according to (2) in [37]:(16)$$\begin{array}{cc} y_{1}\left[t\right] & =K_{1}{d_{0}}^{\gamma }{d_{1}}^{-\gamma }[\sqrt{g_{1}P_{T}}\left(\sqrt{\alpha_{1}}x_{1}\left[t\right]+\hat{j}\sqrt{\alpha_{2}}x_{2}\left[t\right]\right)\\ & +\sqrt{g_{1}P_{T}}\left(\sqrt{\alpha_{4}}x_{4}\left[t\right]+\hat{j}\sqrt{\alpha_{3}}x_{3}\left[t\right]\right)]+A_{1}\\ & +\sqrt{g_{11}\left[t\right]P_{1}}s\left[t\right]+w_{1}\left[t\right], \end{array}$$
where $w_{n}∼CN\left(0,\sigma^{2}\right)$ is the additive white Gaussian noise (AWGN), and s[t] is the signal transmitted by $Bob_{1}$ (received along with the previous transmission) to $Bob_{2}$ in the FD mode, which causes self-interference.

The data rate for the $Bob_{2}$ received by $Bob_{1}$ is given as:(17)$$R_{12}=\log_{2}\left(1+\frac{G_{1}P_{T}\alpha_{2}}{G_{1}\alpha_{3}P_{T}+1}\right),$$
where $G_{n}=K_{n}{d_{0}}^{\gamma }{d_{n}}^{-\gamma }g_{n}/{\sigma_{n}}^{2}$, $K_{n}$ is the path loss constant, $d_{0}$ is the reference distance, $d_{n}$ is the respective distance of the Bob from Alice and ${\sigma_{n}}^{2}$ is the noise variance. For $G_{2}$, $g_{2}$ is the channel power gain, for $G_{12}$, $g_{12}$ is the channel power gain between $Bob_{1}$ and $Bob_{2}$ and so on. The second term of the data rate is the SINR. The data rate of $Bob_{1}$ for detecting its own signal is given as:(18)$$R_{1}=\log_{2}\left(1+\frac{G_{1}P_{T}\alpha_{1}}{G_{1}\alpha_{4}P_{T}+1}\right).$$

Similarly, we can write the data rate for $Bob_{4}$ received by $Bob_{3}$ and the data rate at $Bob_{3}$ of its own signal. The IBFD communication is used only for the priority user $Bob_{2}$; therefore, without the loss of generality a delay τ has been introduced. The total DL received signal by $Bob_{2}$ from Alice and $Bob_{1}$ is given below according to (5) of [37]:(19)$$\begin{array}{cc} y_{2}\left[t\right] & =\sqrt{g_{2}P_{T}}\left(\sqrt{\alpha_{1}}x_{1}\left[t\right]+\hat{j}\sqrt{\alpha_{2}}x_{2}\left[t\right]\right)+\sqrt{g_{2}P_{T}}(\sqrt{\alpha_{3}}x_{3}\left[t\right]\\ & +\hat{j}\sqrt{\alpha_{4}}x_{4}\left[t\right])+\sqrt{g_{12}P_{c}}\hat{j}\left(x_{2}+x_{3}\right)\left[t-\tau \right]+w_{2}\left[t\right]. \end{array}$$

In the above equation, the signal received from $Bob_{1}$ contains only the imaginary part, as $g_{12}A_{2}=0$.

Eve’s received signal can be determined by adding *AN* [14] $A_{1}$ in the second and $A_{2}$ in the third term of (Equation 19). Eve receives signals with high interference due to lack of information about *AN*, the component and R-IBFD. Due to high interference Eve receives a signal with very low SINR and it includes the factor of *AN* which greatly reduces its achievable data rate/capacity $R_{Eve}$ as given in the following equations:(20)$$\begin{array}{cc} y_{e}\left[t\right] & =K_{e}{d_{0}}^{\gamma }{d_{e}}^{-\gamma }[g_{e}\left(\sqrt{P_{T}}\left(\sqrt{\alpha_{1}}x_{1}\left[t\right]+\hat{j}\sqrt{\alpha_{2}}x_{2}\left[t\right]\right)\right)\\ & +g_{e}\left(\sqrt{P_{T}}\left(\sqrt{\alpha_{4}}x_{4}\left[t\right]+\hat{j}\sqrt{\alpha_{3}}x_{3}\left[t\right]\right)+A_{1}\right)]+\\ & K_{1e}{d_{0}}^{\gamma }{d_{1e}}^{-\gamma }g_{1e}\left[t\right]\left(\sqrt{P_{c}}\left(x_{2}+x_{3}\right)\left[t-\tau \right]\hat{j}+A_{2}\right)+w_{e}\left[t\right] \end{array}$$
and
(21)$$\begin{array}{cc} R_{\mathrm{Eve}} & =\log_{2}(1+\frac{G_{1e}P_{1}}{G_{1e}{\left|A_{2}\right|}^{2}+1}+\\ & \frac{G_{e}P_{T}\alpha_{3}}{G_{e}P_{T}\left(\alpha_{1}+\alpha_{2}+\alpha_{4}\right)+G_{e}{\left|A_{1}\right|}^{2}+1}), \end{array}$$
where $G_{n}$ is the total Rayleigh flat fading channel including path loss. Eve is facing interference due to all other signals as shown by $\alpha_{n}$ in the denominator due to absence of component information. The data rate of $Bob_{2}$ and $Bob_{4}$ is sufficiently high, because of no interference of the other road’s users and no $A_{n}$ in the denominator in the equation. Above are the equations for R-IBFD. For other baseline techniques, equations can be written accordingly.

### 4.1. Optimization of Reflective IBFD with ML

The increasing demand for the execution of smart systems in the modern B5G necessitates the use of Machine Learning (ML). It seems with the name that ML is cumbersome and needs intelligence. However, the fact is it needs the intelligence of a machine, not a human. ML is generally helpful for a system with big data and more connected devices. Smart transportation systems also need suitable algorithms for optimization. However, the mathematical techniques may not be suitable for system automation where the machines have to make decisions in a limited period of time with the flexibility of variable data and relevant setup. There are several possible ML algorithms that can be useful, in this paper, MLGA is for the optimization of parameters in the proposed system model.

#### 4.1.1. Machine Learning-Based Genetic Algorithm

MLGA is based on the principles of natural collection and genetics. The problems with mixed continuous-discrete variables, discontinuous variables and non-convex spaces entail optimization for the system design. On-line linear programming methods of optimization can also be useful for the aforementioned problems; though, they will lead to computational expenses and inefficiency. Consequently, the usage of MLGA will be productive. Similar to natural genetic progressions, MLGA is also founded on reproduction, crossover and mutation. Figure 3 characterizes the whole technique that is required for the employment of an MLGA in the corresponding optimization problem [28]. The opted randomly generated parameters are used to obtain the fitness function which is the secrecy capacity in this article. If the fitness function does not achieve the required maximum possible value then the algorithm selects the randomly generated parameters, responsible for the current highest fitness function. The selected parameters are mixed together and exploited to obtain better parameters with crossover. The new parameters are being mutated with some of the old parameters to obtain the best possible fitness function. The process goes on until the best fitness function has been obtained.

The MLGA is a suitable ML technique for an optimization problem. It uses the survival-of-the-fittest principle of nature to maximize the fitness function of the offered problem [38]. In this paper, we used the secrecy capacity as the fitness function of the problem.

#### 4.1.2. Sum Secrecy Capacity Optimization Problems

There are several techniques for system optimization; however, the computation efficiency of using unlimited data leads towards the use of MLGA. The optimization of the sum secrecy capacity requires the secrecy capacity of the individual users affected by the Eve. In this system, we have considered $Bob_{3}$ and $Bob_{4}$ as the affected users.

To make the secrecy capacity better, we optimize it by using MLGA. We deal here with the optimization of the data rate and power coefficients of the system. For optimization of sum secrecy capacity $S_{T}$ of the system, it is considered as the fitness function of the algorithm as shown in Figure 3.

The individual secrecy capacities of each affected vehicle can be calculated as: $S_{n}=R_{n}-R_{ne}$, where n={3,4}. The problem formulation for the secrecy capacity maximization is given as:(22)$$\begin{array}{cc} P1:\;\max:S_{T} & =\;\left(R_{Bob_{2}}-R_{Eve}\right)+\left(R_{Bob_{4}}-R_{Eve}\right). \end{array}$$

The constraints on the power coefficient are:$$C1:\;A\left(\alpha_{1},\cdots ,\alpha_{4}\right)=\sum_{n=1}^{4}\alpha_{n}=1,$$$$C2:\;\alpha_{n}>0,n=\left\{1,\cdots ,4\right\}$$ and the SINR $\zeta_{n}$
$$C3:\;\zeta_{n}>\zeta_{T},n=\left\{1,\cdots ,4\right\}$$where $\zeta_{T}$ is the threshold for the received SINR. In the left hand side of the above equation, there are two round brackets that shows the secrecy capacity of $Bob_{3}$ with the difference between the capacity of $Bob_{2}$ and Eve and $Bob_{4}$.

In the secrecy capacity expression, each term shows the received SINR of the respective $Bob_{n}$ and Eve.

To make the secrecy capacity highly positive we have used the modulation orthogonality and the *AN*. Since Eve does not have the information of the modulation component, therefore it will face more interference and can not decode the information properly. Additionally, the *AN* increases interference of the Eve. A legitimate user is not affected by *AN* because it is the null space of the offered channel. The sum secrecy capacity of the system is given as:(23)$$S_{T}=R_{2}-R_{Eve}+R_{4}-R_{Eve}.$$

The above equation clearly shows the dependence on the capacity of Eve and the affected Bobs. Hence, it is highly subjected to the individual’s received SINR, channel, power, power coefficient and interference. Eve has lower received SINR and hence low $R_{Eve}$ due to high interference as shown in (Equation 21). $R_{2}$ and $R_{4}$ are higher due to less interference for R-IBFD. Therefore, The secrecy capacity will be highly positive.

## 5. Use-Cases of Reflective IBFD

There are a number of use-cases for the proposed R-IBFD communication system. In recent years, exponential growth is observed for dense IoT networks for monitoring critical data that need efficient bandwidth and resource allocation. Internet-of-Energy is an upcoming field including digital smart grids, renewable energy resources, electric vehicle infrastructure, smart metering, smart water system, IoT security and privacy, IoT sensing, data analytic, Software-Defined Networking (SDN)-based IoT fog, industrial mobile IoT, etc. Figure 4 shows the IoT communication system that involves the connected world, home IoT system, smartphones and back-end network systems. Figure also lists several latest offered attacks for an IoT system that need serious consideration. Enabling IoT will increase the number of connected devices which requires the better utilization of the spectrum. IBFD communication can improve the spectral efficiency of the IoT system as compared to HD. R-IBFD is an efficient algorithm that can support the IBFD with minimum interference and prevent the system from security threats like eavesdropping, spyware and man-in-the-middle attacks, as the attacker or eavesdropper cannot download the transmitted signals due to offered system limitations.

R-IBFD is a reliable system and gives a better data rate and therefore can be explored as an Ultra-Reliable and Low Latency Communication (URLLC) in 5G communication. URLLC is needed for the overall IoT system, especially in emergency situations, like natural disasters. Another use-case of R-IBFD communications is in the IoT in health sectors. Health sectors are sensitive areas that need high security and low interference. R-IBFD can provide ideally no interference if a smaller number of users are connected to a certain bandwidth. It will be reliable, secure and safe for the patients exposed to IoT health devices. It is also beneficial for operations for its reliability and speed.

Other use-cases include the use of helicopters and drones in natural disaster situations. A number of natural disasters lead to the destruction of whole cities and villages. In such cases, 5G wireless communication with R-IBFD can help save lives. It can be used for the indication of lives, rescue, provision of food and resources, etc.

In energy sectors, industry 4.0 is moving towards the deployment of the 5G communication system, particularly where the implementation of fibre is not possible. It includes the rural and mountain areas. R-IBFD can fulfil the requirement of fast, reliable and secure communication for remote areas. Such renewable energy resource sites need constant monitoring; however, it is nearly impossible to deploy staff 24 h a day. Security is another important factor that needs to be considered, not only in the energy sectors to prevent power breakdowns but also for communications where life, fraud, destruction risk and confidentiality is involved.

In this paper, a basic communication system is used as an application for the implementation of the proposed R-IBFD. A brief description of how R-IBDF can be implemented for reliable, fast and secure communication is given in the following sections.

## 6. Performance Evaluation

### 6.1. Two-User System

In this section, we discuss the simulated results comparison between the proposed R-IBFD and DF-IBFD cooperative communication system. For the simulation of the systems’ comparison, Rayleigh flat fading channel and 16 QAM has been considered. Other numerical values that have been used for the simulation are given as: $d_{a1}$ = 0.2 m, $d_{12}$ = 0.8 m, γ = 2, $P_{T}$ = 1 W, $P_{1}$ = 0.2 W and $P_{2}$ = 0.8 W. In this paper, Matlab is used as a simulation tool for the comparison between the proposed and the baseline scheme.

For the fair comparison between R-IBFD and DF-IBFD, all selected parameters are the same including AN. The only difference is the interference level during the IBFD mode in both techniques.

Figure 5 shows the comparison for the secrecy outage probability offered by DF-IBFD and R-IBFD. In Figure 5 the transmitted powers of Alice $P_{1}$ and the relay $P_{2}$ are chosen according to the NOMA power allocation strategy. To show the different responses with respect to the number of *K* relays present for selection, we simulated the results for *K* = 1, 2 and 4. SOP result for both algorithms decreases with the increase of SNR. However, R-IBFD outperforms DF-IBFD, due to less interference in the R-IBFD. For the proposed technique, the $\zeta_{1e}$ is approximately equal to 0 due to high interference at Eve. According to the derived equation of outage probability $\zeta_{1e}$, must be greater than zero. Therefore, for simulation purpose we have considered $\zeta_{re}=0.1$ for R-IBFD.

Figure 6 shows the increasing throughput with SNR for DF-IBFD and R-IBFD schemes. To demonstrate a diverse scenario with respect to the number of relays *K*, this paper presents the results for *K* = 1, 2 and 4. Throughput of R-IBFD is higher due to its low SOP as compared to DF-IBFD algorithm. It can be seen from the figures that with K=4 the throughput is optimum. The level of throughput depends on the relay(s) selection. Additionally, the proposed R-IBFD with K=2 performs better than DF-IBFD with K=4. Minimum interference in R-IBFD makes it reliable. It is due to the fact that in R-IBFD, the message signal uses half bandwidth for the transmission comparatively.

### 6.2. Four-User System

This section evaluates the proposed R-IBFD system performance. For the simulation 16-QAM modulation, Rayleigh fading channel is used and total transmission power $P_{T}$ = 1 W is used. Secrecy capacity comparison with baseline techniques is evaluated, MLGA is also used for the optimization of secrecy capacity.

The baseline comparative schemes are NOMA R-IBFD without *AN*, Reflective HD (R-HD), R-HD with *AN* (RA-HD) and R-IBFD with *AN* (RA-IBFD).

Figure 7 shows the simulated result for secrecy capacity with a four-user system and a passive Eve. The comparison of RA-IBFD and R-IBFD with R-HD, RA-HD, and NOMA shows that the proposed RA-IBFD outperforms all other implementations. It is due to the fact (Equation 21) that the Eve faces a high amount of interference due to the addition of *AN*, no CSI information of the Bobs and the desired component of the particular Bob. Therefore, it leads to a significant gain of RA-IBFD.

The high interference on the Eve node increases the difference between the capacities of the respective Bob and the Eve, which leads to the increase of secrecy capacity. Figure 8 shows the simulated result of RA-IBFD with and without MLGA. For MLGA, the power coefficients are generated randomly. For each $Bob_{i}$, four random numbers have been generated as the initialized population with respect to distance and with the sum equal to 1. The secrecy capacity is used as the fitness function of the MLGA. The optimized secrecy capacity has been simulated using the MLGA method as shown in Figure 3. The result shows that MLGA further enhances the performance of RA-IBFD due to the optimized parameters subjected to the optimized level of RA-IBFD secrecy capacity. Therefore, the use of MLGA makes improves the secrecy capacity. For a limited number of users as chosen in this article, it is easy to measure system parameters statistically. However, in a system with a high number of users, machine learning is a necessity for a wireless communication system with high data.

## 7. Conclusions

IBFD is a SE technology that can meet the target of the high data demand in the 6G era, where a relay or an existing user assists a far user in an IBFD mode. The exploration of proposed R-IBFD and RA-IBFD has shown the enhancement of the secrecy capacity, along with the SE and reduced interference.

The system fulfils the requirement of high data rate using proposed R-IBFD and RA-IBFD. The high data rate or high capacity will lead towards the management of big data and high secrecy capacity. The ML is a part of the modern systems of wireless communication, it can be used for multiple purposes where it improved the system’s performance without altering and disturbing its surrounding. This paper is an extension of a conference paper [36]. In this paper, an SOP equation is derived for a selection of relay in R-IBFD, MLGA is used for the optimization of power coefficients subjected to secrecy capacity of the given system, this will be particularly helpful for dealing with larger number of users and data. Overall, R-IBFD is an efficient system that provides a better transmission data rate and enhances security. Other ML and statistical algorithms can also be used to improve the performance of the proposed algorithm [39], which will be further explored as a part of future research of the proposed algorithm. Alongside the simulation results, use cases of the proposed algorithm are also given in this paper. Further research is highly required for the flexibility of systems with several real-life 6G communication environment that needs to be explored for other security reasons including active attacks during public holidays.

## Figures and Tables

**Figure 1 sensors-22-02508-f001:**
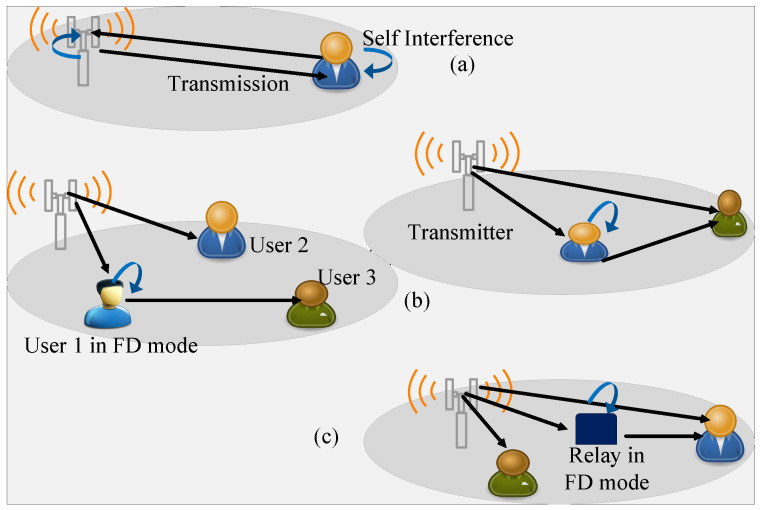
Three different ways of combining FD with NOMA. (**a**) When transmission is in both downlink and uplink. (**b**) D2D communication with a direct and no direct link between transmitter and the distant user. (**c**) The use of a FD mode relay for assisting the distant user.

**Figure 2 sensors-22-02508-f002:**
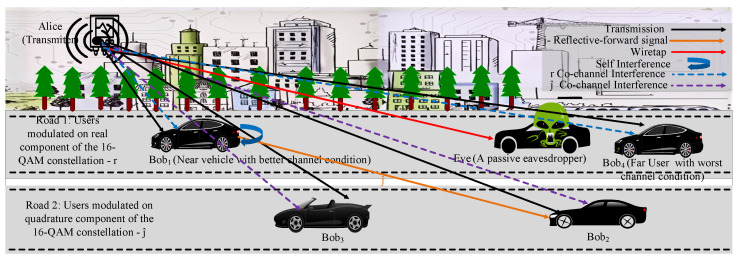
The system model with an Alice, four Bobs and an Eve. Each Bob is modulated differently following the proposed algorithm, for the reduced NOMA interference. $Bob_{1}$ shows the IBFD nature and assists $Bob_{3}$ with an R-IBFD transmission. The given system involves self and co-channel interference.

**Figure 3 sensors-22-02508-f003:**
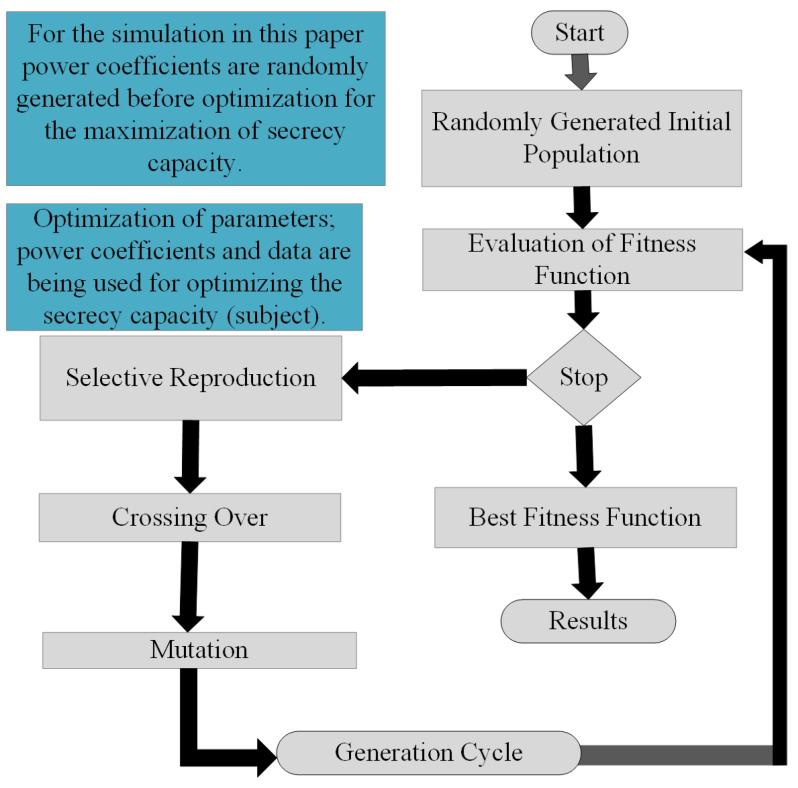
Flow chart description of genetic algorithm based ML.

**Figure 4 sensors-22-02508-f004:**
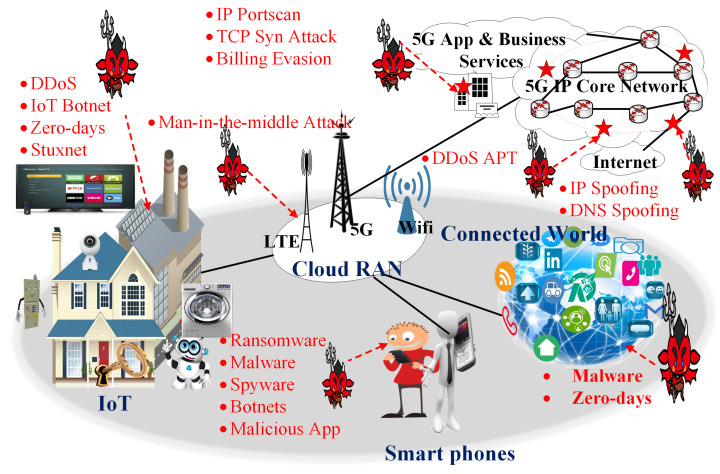
Possible cyber-security attacks in a 5G wireless communication system.

**Figure 5 sensors-22-02508-f005:**
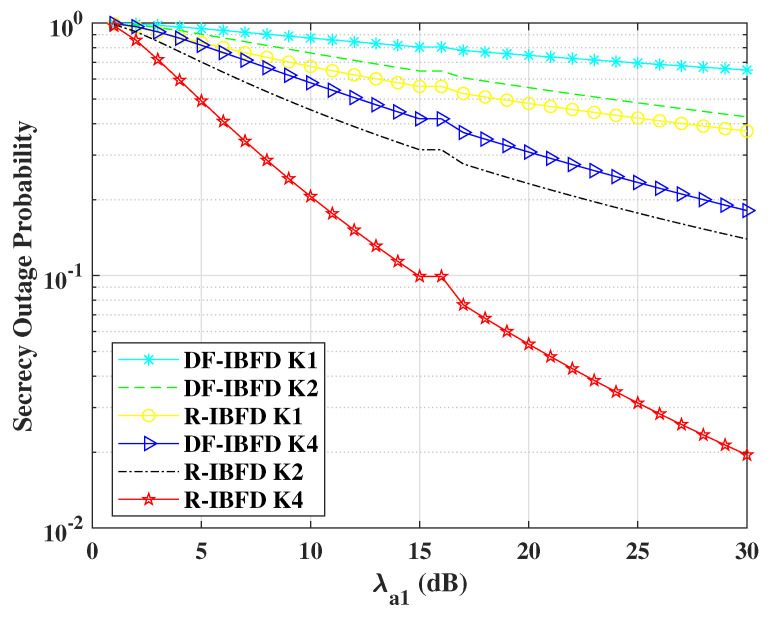
Secrecy outage probability comparison for Reflective In-Band Full-Duplex (R-IBFD) and Decode-and-Forward In-Band Full-Duplex (DF-IBFD) with $C_{th}=1$, $\zeta_{12}=\zeta_{11}=$ 12 dB, *K* = 1, 2 and 4 and $\zeta_{1e}$ is calculated with respect to Rayleigh flat fading and respective interference.

**Figure 6 sensors-22-02508-f006:**
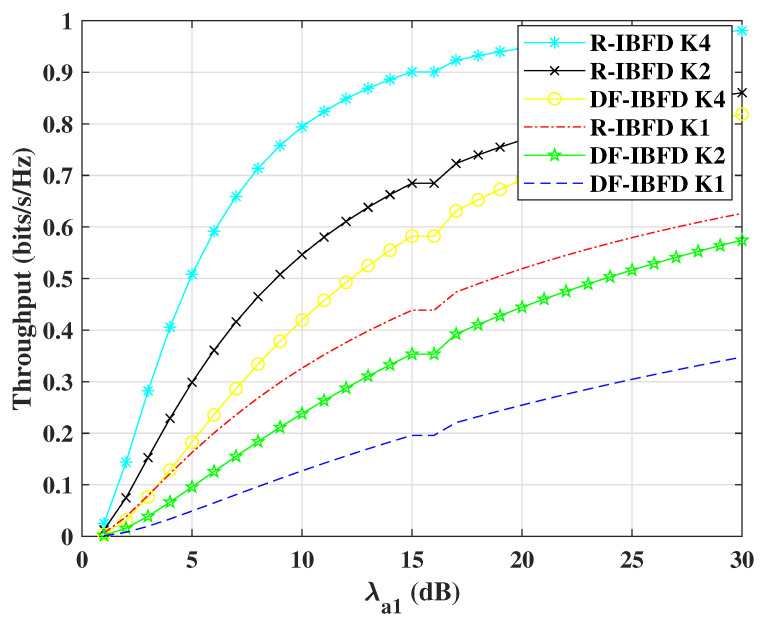
Throughput comparison for Reflective In-Band Full-Duplex (R-IBFD) and Decode-and-Forward IBFD (DF-IBFD) with $C_{th}=1$, $\zeta_{12}=\zeta_{11}=$ 12 dB, *K* = 1, 2 and 4 and $\zeta_{1e}$ is calculated with respect to Rayleigh flat fading and respective interference.

**Figure 7 sensors-22-02508-f007:**
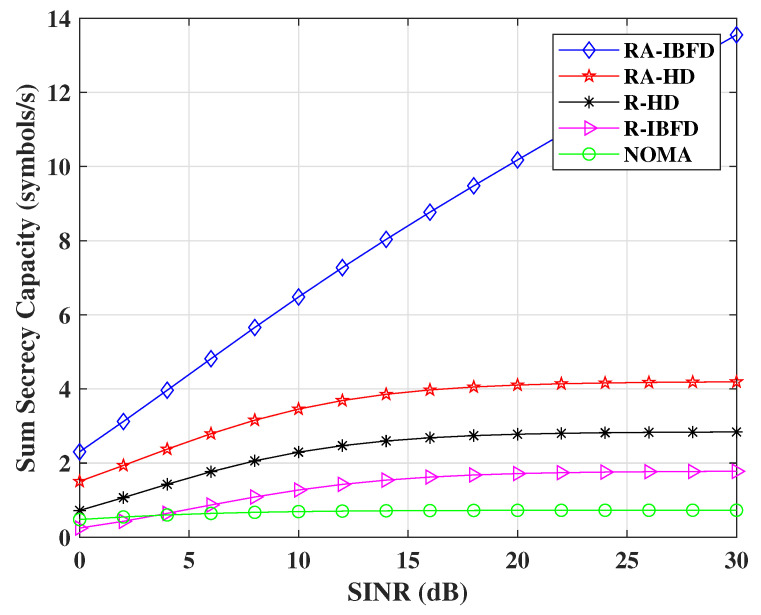
Sum secrecy capacity comparison for Proposed RA-IBFD and R-IBFD with R-HD, RA-HD and NOMA.

**Figure 8 sensors-22-02508-f008:**
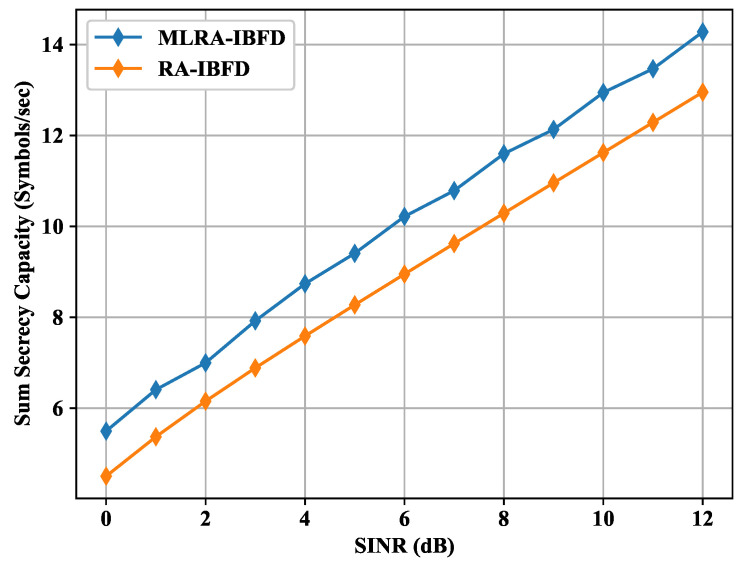
Sum secrecy capacity comparison for Proposed RA-IBFD with and without MLGA.

**Table 1 sensors-22-02508-t001:** Comparison study of existing papers that include NOMA with OFDMA.

Citation	Theme	Discussed Technologies	Target
[4]	C-RAN-based FD-NOMA	Relay-based cooperative communication and D2D	Performance comparison and measurements
[5]	Applications of FD-NOMA	Cellular, relay and cognitive radio	Comparison and discussion
[6]	QoS with 5G spectrum	FD, massive MIMO, NOMA, SWIPT	Performance evaluation and comparison between the technologies
[7]	Combination of potential technologies	massive MIMO, mmWave, FD, NOMA, carrier aggregation, CR, and network ultra-densification	Combined coupling factors, problems and possible solutions for the existing literature combination
[3]	Spectrum Sharing in 5G	D2D, in-band FD, NOMA, and LTE	Discussion of research methodologies and challenges in 5G networks

## Data Availability

This article used randomly generated data for MLGA. Python is used as a tool for the simulation of MLGA, the basic command for randomly generated sequence of Python is used to generate data.

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
