# Peer review of "Interference Management with Reflective In-Band Full-Duplex NOMA for Secure 6G Wireless Communication Systems"

_sensors, 2022, doi:10.3390/s22072508_

Round 1

Reviewer 1 Report

This paper studied the secure capacity of a full-duplex NOMA system. The major problems of this

paper are given as follows:

  1. This paper spent a lot of space to introduce the development for NOMA and Full-duplex. It led to that this paper seems like a summary or review, rather than a regular paper in journal. Thus, the reviewer confuses the motivation of this paper.
  2. The proposed system model in section 4 lack of novelty. In addition, the reviewer cannot find the universality of this system model. It just a special case in full-duplex NOMA network. Thus, please highlight the them in this paper.
  3. It is very strange that the symbol ‘g_n’is defined as channel condition in pervious. However, in next, ‘g_n’is defined as the channel power gain. Please integrate them.
  4. This paper adopted genetic algorithm to solve Problem P1. The genetic algorithm is untimely, and it can be used in the most optimization problems. The authors need to clarify why consider the genetic algorithm rather than other algorithms.
  5. The reviewer doubts the performance of genetic algorithm. Please compare the time complexity between the MLGA and the conventional optimized algorithms.

Reviewer 2 Report

This paper proposes a scheme that involves full-duplex communications and NOMA technologies. This is a timely topic; however, there are some issues that should be clarified before further considerations.

1) In the abstract, it is mentioned that the proposed scheme: "provides efficient use of spectrum with better system parameters including Spectral Efficiency (SE), Energy Efficiency (EE), throughput, data rate, Quality of Service (QoS), ..." However, the authors have not considered EE metrics in the paper. Please elaborate more on this by providing specific examples.

2) The considered optimization problem in the paper is not well formulated from mathematical point of view. The authors should clearly mention the optimization parameters and discuss why this problem is difficult to be solved.

3) It is not clear why the authors considered only ML techniques to optimize the parameters. The authors should also provide an analytical solution and then compare the performance of the ML-based algorithm with the analytical approach in terms of optimality and complexity. 

4) It seems that the authors only considered the sum-secrecy rate for a system with 2 users. If it is the case, it should be clearly mentioned in the abstract and introduction.

5) It is interesting to verify how this scheme performs if we maximize the minimum rate of users. The authors should elaborate on it.

6) There writing of the paper should be improved.

Author Response

Please find the attached file for reviewer 2.

Round 2

Reviewer 1 Report

The authors have successfully addressed all the comments, and thus this paper now be accepted.

Author Response

Thank you so much for your time, support and accepting the paper for submission. Highly appreciated.

Reviewer 2 Report

The contribution of the paper is not clear. The paper does not propose any machine learning technique and/or any other statistical optimization algorithm. The optimization is completely conducted based on the simulations, which is very heuristic. The discussion on NOMA in the paper is not precise. It is also very difficult to follow how the authors addressed my previous issues in the new manuscript. Overall, the paper must be improved significantly.

Author Response

We are sorry to know that our answers were not helpful to the reviewer. Therefore, we have reviewed our answers which were sent in response previously and attached again in the beginning so that reviewer can have a clear understanding.

The contribution of the paper includes a cooperative communication technique for NOMA when a far user needs support from a near user, the near user can forward the signal in the in-band full duplex mode ideally without interference. It is the novel technique which claims to have ideally no interference during in-band full duplex mode for a NOMA system. The technique is given a name of reflective in-band full duplex (R-IBFD).

For reviewer and other readers, we have elaborated the definition of in-band full duplex, NOMA, and corresponding interference in the paper.

The proposed technique (R-IBFD) is further explored for system’s physical layer security in the paper. This is have exploited using artificial noise and the orthogonal separation involved in the proposed idea of R-IBFD. For that we have derived secrecy outage probability statistically and then simulated it using the simulation tool. For the secrecy capacity we have given the basic equations involved and optimized is with simulation of ML.

As seen in figure 8, we have compared the analytical secrecy capacity with the machine learning and the text of explanation is highlighted in the blue colour.

Reviewer seems confused about our use of machine learning optimization. We have used machine learning to show the potential optimization of the proposed R-IBFD security. We have not proposed any other statistical or ML optimization in this paper. This is not proposed in this paper. The proposed idea is elaborated in the paper and is elaborated in the previous paragraphs. Machine learning genetic algorithm (MLGA) does not include any statistical calculations, it is done only using simulations. The reason is given in the paper in the respective section. Also, reviewer can use the following references to see that MLGA does not require any statistical information.

To satisfy the requirement of reviewer, we added some mathematical analyses of the proposed technique for secrecy outage probability.

  • Yin, W. Chenggong, M. Kai, B. Kuanxin, “A NOMA Power Allocation Strategy Based on Genetic Algorithm,” in Communications, Signal Processing, and Systems, vol. 571, pp. 2182-2190, 2019.
  • Adnan, R. Green, and E. Hines, “A neuro-genetic hybrid algorithm utilizing outdoors LOS optical wireless channels”, Fourth International Conference on Computational Intelligence, Communication Systems and Networks, Jul. 2012.

  • Kaustubh, and R. Bhattacharjee, “Bit error rate performance of genetic algorithm optimized WDM systems”, IFIP International Conference on Wireless and Optical Communications Networks, IEEE, Apr. 2006.
  • Sheng, et al. “Decoding algorithm of LDPC codes based on genetic algorithm”,(2013): 2-14.
  • Sujee and K. E. Kannammal, “Energy efficient adaptive clustering protocol based on genetic algorithm and genetic algorithm inter cluster communication for wireless sensor networks”, International Conference on Computer Communication and Informatics (ICCCI) 2017.

  • Sajid, A. W. Matin and O. Islam, “Genetic algorithm for hierarchical wireless sensor networks”, Journal Of Networks vol. 2, no. 5, Sep. 2007.
  • D. Higgins, R. J. Green, M. S. Leeson, EL. Hines, “Genetic algorithm optimisation of the SNR for indoor optical wireless communication systems”, 12th International Conference on Transparent Optical Networks. IEEE, 2010.

  • B. Ranjan, “Vivaldi antenna for UWB communications: Design, modelling and analysis of Vivaldi Antenna with genetic algorithm”, International Conference on Control, Computing, Communication and Materials (ICCCCM), IEEE, Oct. 2016

A thorough review on NOMA is given in the introduction section . We have now ighlighted that in blue colour.

Round 3

Reviewer 2 Report

The authors have improve the representation and quality of the paper. I do not have any further comment.